# Predictors for carotid and femoral artery intima-media thickness in a non-diabetic sleep clinic cohort

Christopher Lambeth[1,2], Rita Perri[1,2], Sharon Lee[1,2], Manisha Verma[1,2], Nicole Campbell-Rogers[3], George Larcos[3,4], Karen Byth[5], Kristina Kairaitis[1,2,4], Terence Amis[1,2,4]*, John Wheatley[1,2,4]

1 Ludwig Engel Centre for Respiratory Research, The Westmead Institute for Medical Research, Westmead, Sydney, New South Wales, Australia, 2 Department of Respiratory and Sleep Medicine, Westmead Hospital, Westmead, Sydney, New South Wales, Australia, 3 Department of Nuclear Medicine and Ultrasound, Westmead Hospital, Westmead, Sydney, New South Wales, Australia, 4 Sydney Medical School, University of Sydney, Sydney, New South Wales, Australia, 5 Research and Education Network, Western Sydney Local Health District, Westmead Hospital, Westmead, Sydney, New South Wales, Australia

* terry.amis@sydney.edu.au

**Data Availability Statement:** The study's minimal underlying data set has been uploaded as Supporting information.(compressed files).

## Abstract

### Introduction

The impact of sleep disordered breathing (SDB) on arterial intima-media thickness (IMT), a surrogate measure for cardiovascular disease, remains uncertain, in part because of the potential for non-SDB vascular risk factor interactions. In the present study, we determined predictors for common carotid (CCA) and femoral (CFA) artery IMT in an adult, sleep clinic cohort where non-SDB vascular risk factors (particularly diabetes) were eliminated or controlled.

### Methods

We recruited 296 participants for polysomnography (standard SDB severity metrics) and CCA/CFA ultrasound examinations, followed by a 12 month vascular risk factor minimisation (RFM) and continuous positive pressure (CPAP) intervention for participants with a range of SDB severity (RFM Sub-Group, n = 157; apnea hyponea index [AHI]: 14.7 (7.2–33.2), median [IQR]). Univariable and multivariable linear regression models determined independent predictors for IMT. Linear mixed effects modelling determined independent predictors for IMT change across the intervention study. P<0.05 was considered significant.

### Results

Age, systolic blood pressure and waist:hip ratio were identified as non-SDB predictive factors for CCA IMT and age, weight and total cholesterol:HDL ratio for CFA IMT. No SDB severity metric emerged as an independent predictor for either CCA or CFA IMT, except in the RFM Sub-Group, where a 2-fold increase in AHI predicted a 2.4% increase in CFA IMT. Across the intervention study, CCA IMT decreased in those who lost weight, but there was

**Funding:** This study was funded by two National Health and Medical Research Council (NHMRC) grants (632597; APP1024440), www.nhmrc.gov.au. The funders had no role in study design, data collection and analysis, decision to publish, or preparation of the manuscript.

**Competing interests:** The authors have declared that no competing interests exist.

no CPAP use interaction. CFA IMT, however, decreased by 12.9% (95%CI 6.8, 18.7%, p = 0.001) in those participants who both lost weight and used CPAP > = 4hours/night.

## Conclusion

We conclude that SDB severity has little impact on CCA IMT values when non-SDB vascular risk factors are minimised or not present. This is the first study, however, to suggest a potential linkage between SDB severity and CFA IMT values.

## Trial registration

Australian New Zealand Clinical Trials Registry, ACTRN12611000250932 and ACTRN12620000694910.

## Introduction

Sleep disordered breathing (SDB) is a common consequence of increased upper airway resistance during sleep [1]. SDB is associated with a number of different comorbidities, including both cardio- and cerebrovascular disease [2–4]. While the mechanisms underlying these associations remain poorly understood, epidemiological studies suggest a strong relationship between obstructive sleep apnoea (OSA) and stroke [5, 6]. People with moderate to severe OSA were at substantially increased risk of stroke in the Busselton Sleep Cohort [5], while stroke risk for men in the Sleep Heart Health Study increased by 6% for each unit increase in apnoea-hypopnea index (AHI) between 5–25 events/hr [6].

A number of plausible mechanisms have been proposed linking SDB and macrovascular disease, including intermittent hypoxia, intrathoracic pressure swings and sleep fragmentation [4, 7]. But despite the plausibility of a mechanistic role for SDB, the evidence is not clear. The SAVE trial failed to support a causal relationship between OSA and cardiovascular disease showing no significant reduction of cardiovascular or cerebrovascular events when OSA was treated with continuous positive airway pressure (CPAP) [8]. It is possible, however, that more sensitive measures of cardiovascular benefit were overlooked.

One such measure may be the carotid intima-media thickness (IMT), which is a simple and non-invasive ultrasonographic method to evaluate subclinical cardiovascular disease and early atherosclerosis, and a predictor of future cerebral and cardiovascular events [9, 10]. The potential confounding influence of known macro-vascular disease risk factors (RF) makes it difficult to clarify interactions with OSA. Further, the high prevalence of these RFs in sleep clinic cohorts means an independent effect for SDB is likely to be difficult to detect. This issue was highlighted in a recent meta-analysis of 18 studies comparing carotid IMT for individuals with and without OSA [11]. The authors found strong heterogeneity in study results suggesting inconsistency in adjustment for confounding factors. Moreover, in a sub-group analysis, better matching of confounding variables uncovered an increase in carotid IMT for OSA patients compared to controls, and a significant correlation between AHI and carotid IMT (r = 0.389; P<0.001).

While the focus of SDB research has been on associations with the carotid IMT, it has been suggested recently that the femoral artery IMT may be a more sensitive marker of cardiovascular risk [12, 13]. While femoral and carotid IMT are strongly correlated with each other, the RFs associated with each differ [14]. It has also been suggested that local haemodynamic

factors may have a larger impact on the development of atherosclerosis in the carotid artery than in the femoral artery and, thus, systemic RFs may be more important for the femoral artery [15]. Thus, there is appeal in exploring the relationship between SDB and femoral IMT.

CPAP therapy is first line treatment for OSA, however, reported effects of CPAP on carotid IMT are inconsistent [16–20]. While there are some methodological concerns about these studies, a recent meta-analysis found CPAP had no overall effect on carotid IMT in OSA patients [21], but there was a decrease in carotid IMT in a sub-group of patients with more severe OSA and longer term CPAP use. It is possible that studying the femoral IMT, as a surrogate measure of vascular health, may be advantageous in exploring the effects of CPAP.

Accordingly, the aims of the present study were to determine predictors for carotid and femoral IMT in a sleep clinic cohort and determine their relationships with SDB severity. In our study design, we attempted to minimise interactions with classical non-SDB vascular disease RFs by excluding individuals with diabetes. We then combined objective ultrasound measurements of IMT with gold-standard, in-laboratory, polysomnography (PSG) for assessment of SDB. The study involved a cross-sectional analysis in which statistical modelling was used to search for significant predictive variables based on improvements in estimates of explained variance. We also included a longitudinal (12-month), intervention-based sub-group study, aimed at complementing the main cross-sectional study results by testing the impact on IMT of minimising SDB using CPAP when superimposed on a background of medically supervised vascular disease RF stabilisation.

## Methods

### Protocol and registration

The study described in this manuscript was not registered as a clinical trial before enrolment of participants because it was originally designed as an observational study, however, the authors now confirm that all related trials included in the analysis described in this manuscript have now been registered with the Australian New Zealand Clinical Trials Registry (ANZCTR; Trial Ids: ACTRN12611000250932 and ACTRN12620000694910).

### Study participants

We recruited 373 adults all over the age of 35 years, with no known history of carotid artery disease/surgery, all referred to a research sleep clinic between 5th September 2011 and 25th July 2016, for investigation of potential SDB and all evaluated by the same senior sleep physician (Author: KK). Participants were screened for diabetes and 42 were subsequently excluded on the basis of either a history of diabetes or a fasting blood sugar level (BSL fasting) $\geq 7$ mmol/L, while 35 other participants either withdrew from the study or were excluded for other reasons (e.g. failure to attend study visits). The remaining 296 participants formed the study "Main Group".

Participants were then invited to participate in a 12-month duration RF minimisation intervention (RFM) sub-study conducted between 13th September 2011 and 6th December 2016. A feature of this sub-study was that individuals encompassing a wide range of SDB severity levels were invited to participate, with the goal of recruiting participants with a range of SDB severity for analysis purposes. Enrichment of this sub-group with individuals across the SDB severity spectrum meant that some were included in the sub-study who wouldn't have been considered for CPAP on clinical criteria. A total of 157 participants ("RFM Sub-Group") agreed to enter this phase of the study and committed to regular monitoring and 12 month follow-up data collection. Fig 1 shows a flow diagram of the study design. Informed written consent was

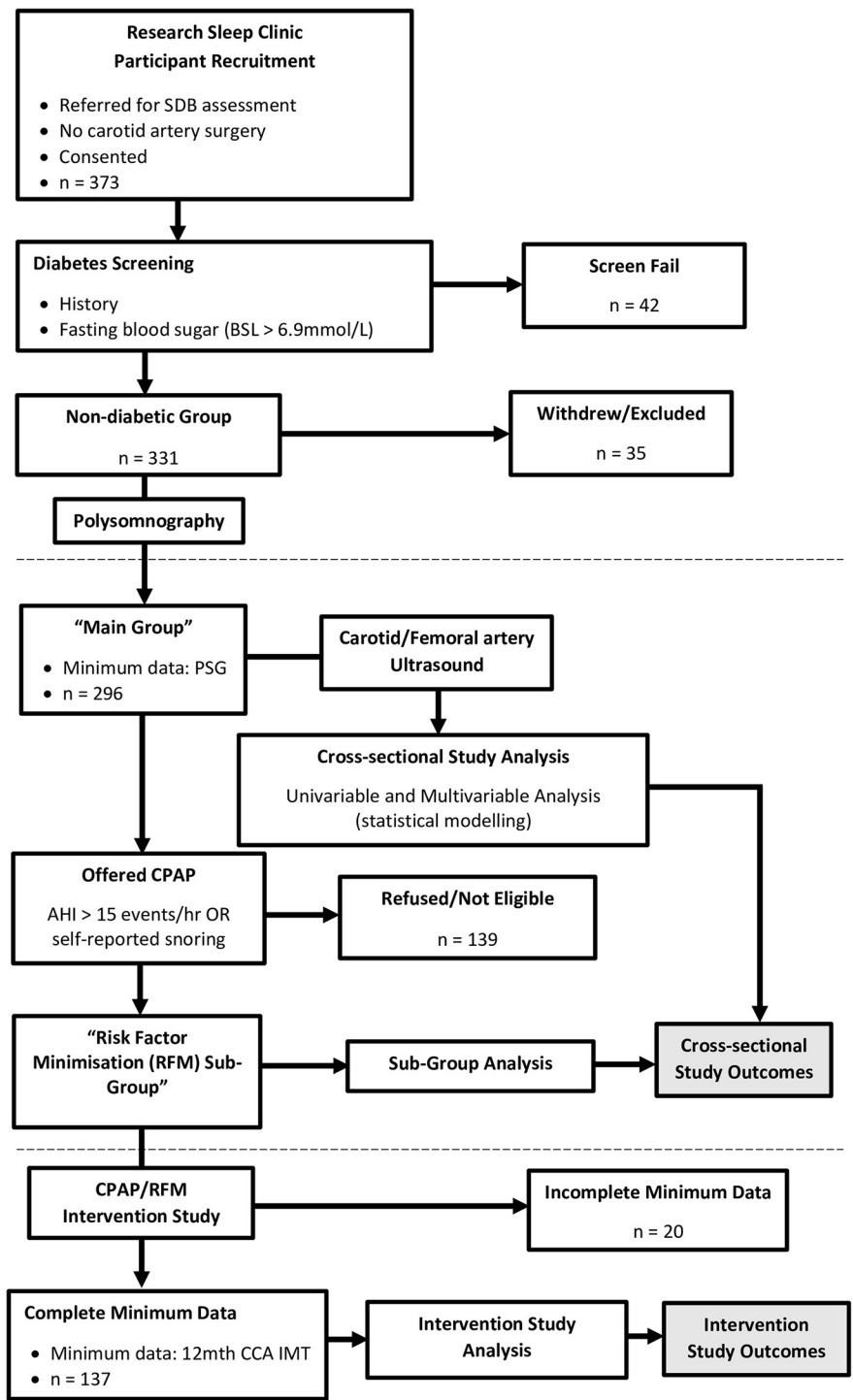

**Fig 1. Study flow diagram.** Study flow diagram showing movement of participants and data through the recruitment and screening phases, the "Main Group" and "Risk Factor Minimisatation [RFM] Sub-Group" cross-sectional analyses, and the intervention study. SDB = sleep disordered breathing; BSL = blood sugar level; PSG = polysomnography; CPAP = continuous positive airway pressure; AHI = apnea hypopnea index; CCA IMT = common carotid artery intima-media thickness.

obtained, and the protocol was approved by the Sydney West Area Health Service Human Research Ethics Committee.

## Study protocol

**Cross-sectional study.** The Main Group study protocol included measurement of anthropometric data (height, weight, body mass index [BMI], waist circumference, hip circumference, waist hip ratio [WHR], neck circumference), detailed medical history (history of hypertension and hypercholesterolaemia), smoking history (current plus past smokers versus never smoked), measurement of blood pressure (BP), fasting blood tests (total cholesterol, high-density lipoprotein [HDL] cholesterol, glucose) and assessment of cardiovascular risk (Framingham Risk Score). Carotid and femoral artery IMT were measured using B-mode ultrasound [22, 23], while sleep disordered breathing was assessed via in-laboratory overnight standard polysomnography (PSG) [24].

## Polysomnography

Sleep disordered breathing was assessed by standard polysomnography [24]. Nasal pressure was used to measure airflow for scoring hypopneas. Pulse oximetry recorded blood oxygen saturation ($SpO_2$). Studies were scored, using Compumedics Profusion PSG4 software (Compumedics Limited; Abbotsford, Victoria, Australia) according to standard guidelines [24, 25] and apnoea-hypopnea index (AHI), respiratory disturbance index (RDI), oxygen desaturation index (ODI >3%), arousal index (AI), the percentage of sleep time spent with oxygen saturation below 90% ($SpO_2$<90%) and the lowest $SpO_2$ during NREM and REM sleep were all calculated.

OSA severity was categorised by AHI as normal (AHI<5 events/hr), mild ($5 \leq$AHI<15 events/hr), moderate ($15 \leq$AHI<30 events/hr) or severe (AHI$\geq$30 events/hr). For analysis, OSA was also categorised as either non-severe (AHI<30 events/hr) or severe (AHI$\geq$30 events/hr).

In the RFM Sub-Group, snoring sounds were recorded using a unidirectional sound level meter (Rion) suspended 60 cm above the subject. Snores were counted manually and expressed as snores/hr of sleep.

## Carotid & femoral ultrasound

Carotid and common femoral artery B-mode ultrasounds were obtained using a SonoSite M-Turbo ultrasound system, using either a HFL38x 13–6 MHz or an L38x 10–5 MHz transducer (Fujifilm SonoSite, Inc; Bothell, WA, USA). Participants lay supine with their upper body slightly raised, neck extended and turned contralateral to the side being examined. Several longitudinal images were obtained of the common carotid artery (CCA), with the carotid bulb visible at the far left of the image and at least 20mm of the CCA intima-media layer proximal to the bulb. Similar images were obtained of the common femoral artery (CFA) with the participant supine and their leg slightly abducted, with the femoral bifurcation at the far right of the image and at least 20mm of CFA intima-media layer proximal to the bifurcation. The images were stored for later offline analysis.

A single experienced sonographer assessed the images and made IMT measurements using the SonoCalc IMT software (Fujifilm Sonosite, Inc) trace methodology with at least three reference points. Measurements were made on the posterior wall of the CCA and CFA over a 10mm segment, with measurements in the CCA acquired in a 10–20 mm segment inferior to the carotid bulb. The software calculated the mean IMT. All IMT measurements were acquired three times for each location (left and right CCA, left and right CFA) and these three values

were then averaged to give a single measurement for each site. Left and right values were then averaged to give overall CCA and CFA IMT values.

**Longitudinal (RFM) intervention study.** After recruitment to the intervention study, participants (RFM Sub-Group) with hypertension or hypercholesterolaemia were prescribed appropriate medication and had a 1-month run-in period to allow stabilisation of their blood pressure and lipids before undergoing a second PSG to determine the appropriate CPAP pressure [26] for treatment of their sleep disordered breathing. They were assigned a CPAP machine (ResMed S9; ResMed Ltd., Bella Vista, NSW, Australia; or Respironics System One; Philips Respironics, Murrysville, PA, USA) and fitted with an appropriate mask. Over the next 12-months, participants received personalised support, including monthly medical monitoring of their health and medication status, counselling to encourage continued nightly CPAP use, and regular downloads of their CPAP usage. At the conclusion of the study, repeat measures of anthropometrics, blood pressure and carotid/femoral IMT were obtained and data were analysed for CPAP compliance (hours of use).

## Statistical analysis

All statistical analyses were performed using SPSS version 24.0 (IBM SPSS Statistics for Windows, Version 24.0. Armonk, NY: IBM Corp.). Except where stated, P<0.05 was considered significant.

Note: The study described in this manuscript deviates from the registered clinical trials (ACTRN12611000250932 and ACTRN12620000694910). Both trials were conducted simultaneously, used the same methodologies, and addressed the same outcome variables, but with study cohorts focused on different degrees of sleep disordered breathing severity. However, because there was considerable overlap in the range of measured sleep disordered breathing severities between the two study cohorts they were combined post hoc into one overarching analysis.

**Cross-sectional study.** Group data were summarised using mean (± standard deviation) or median (interquartile range), where appropriate, for continuous variables, and frequency and percentage for categorical variables. Since many continuous variables were skewed, Spearman rank correlation (r) was used to quantify the pairwise associations between them. Linear regression was used to define the age relationship for both CCA and CFA IMT. Differences between the RFM Minimisation Sub-Group and those Main Group participants not included in the RFM study were tested using two-samples t-tests or Mann-Whitney U tests where appropriate for continuous variables, and chi-square tests for categorical variables.

Our analysis strategy first identified non-sleep related variables predictive of IMT operating within the data set (base models) and then assessed the impact of adding individual SDB variables to the base models as described below. IMT and the continuous SDB variable values were skewed. These data were log transformed (after adding 1 to SDB variables to deal with potential zero values) to approximate Normality and to stabilise the variance prior to analysis. Model parameter estimates for these variables were back transformed and relationships reported in terms of relative (rather than absolute) change on the original scale of measurement. The models based on relative change relationships were considered biologically plausible and more closely adhered to underlying statistical assumptions. Diagnostic residual plots (including residuals versus fitted values and Normal probability plots of the residuals) were used to check the adequacy of the fitted models.

Backward stepwise multiple linear regression (variables removed at p>0.1) was used to identify independent non-sleep variable predictors from amongst candidate variables that were identified in two ways: 1) based on published models for IMT [27–29]; and 2)

demonstrated significant (p<0.05) univariable associations with IMT within the cross-sectional study data. In addition, a 'study group' factor ('included' in RFM Sub-Group versus 'not included') was included to determine whether the same set of non-sleep variables were operating in the RFM sub-group. Collinear variables were identified by collinearity tests–variance inflation factor (VIF>10), condition index (>30) and variance proportion (>0.5)–and were removed based on weaker univariate associations with the dependent variable. Identified candidate variables were then entered into the backward stepwise regression and a base model established for each dependent variable (ln(CCA IMT) or ln(CFA IMT) base model). As the study group factor was not significant in these models, the same base models were used for analysis of both the Main Group and RFM Sub-Group.

SDB variables, AHI category and severe/non-severe OSA) were added to the base models individually to assess the effect of each SDB variable after adjusting for the base model. These models were also checked for collinearity, as above. The base-model-adjusted parameter estimate and its standard error (SE) are presented for each SDB variable along with the incremental change (SBD model-Base Model) in $R^2$ ($\Delta R^2$) to determine whether the addition of the SDB variable to the base model improved overall model performance. An interaction term [SDB variable x study group] was added to test whether the base-model-adjusted SDB effect differed between those included in the RFM Sub-Group and those not included. If a significant SDB variable x study group] interaction was detected, the base-model-adjusted SDB parameter was also estimated in only the RFM Sub-Group.

**RFM intervention study.**   Paired t tests were used to test for within participant change in log transformed IMT values over 12 months and results expressed as percentage change in IMT from baseline with 95% confidence interval (95%CI). For each variable in the cross-sectional study base models, the within participant change over 12 months was calculated as: $\Delta$variable = (variable$_{12month}$−variable$_{baseline}$). Spearman rank correlation was used to quantify the association between the change in ln(IMT) and each $\Delta$variable.

Linear mixed effects (LME) models were used to examine the effect of CPAP use on ln (IMT) values over the 12 month RFM study, either as a continuous variable (average hours/day) or as a dichotomous variable (<4hrs/day vs > = 4hrs/day). In these models, subject was used as the group identifier. The 'Visit' (1 = baseline, 2 = 12 months) variable was fitted as both a random effect with unstructured covariance matrix and as a fixed effect. The covariates from the relevant base model were added, along with CPAP use and its interaction with Visit, as fixed effects. A significant (CPAPxVisit) interaction term was interpreted as evidence that CPAP use influenced the within participant change in ln(IMT) from baseline to 12 months. Diagnostic plots were used to assess the adequacy of the fitted models. They included scatter plots of standardized residuals by fitted values and Normal probablity plots of residuals and of estimated random effects to check the normality assumption for the within-subject errors and for the random effects.

## Results

### Cross-sectional study

Table 1 shows Main Group anthropometrics, Framingham Cardiovascular Risk values, polysomnography and ultrasound data. Additional anthropometrics, medical history, cholesterol and blood sugar levels can be found in S1 Table in S1 File. Overall, the Main Group has 55% males, with 82% having low-intermediate cardiovascular risk levels and a wide range of SDB, skewed towards the lower end of the AHI spectrum.

Table 2 shows the same data for the RFM Sub-Group which had a significantly higher proportion of males, increased body size metrics (height, weight, neck circumference and waist:

**Table 1. Main group demographics and sleep disordered breathing status.** Demographics (A), cardiovascular risk status (B), IMT and SDB status (C) of the Main Group, presented as mean ± SD, median (IQR) or frequency, where appropriate, plus range. See text for abbreviations.

**A**

|  | N | Mean ± SD | Range |
|---|---|---|---|
| Age (yrs) | 296 | 57.9 ± 9.3 | 35–79 |
| Height (cm) | 295 | 167.5 ± 9.7 | 144.0–191.5 |
| Weight (kg) | 295 | 88.9 ± 20.9 | 41.6–186.3 |
| BMI (kg/m$^2$) | 295 | 31.7 ± 6.9 | 17.1–64.5 |
| Neck circumference (cm) | 293 | 39.7 ± 4.5 | 30.0–56.5 |
| Waist:hip ratio | 293 | 0.94 ± 0.08 | 0.52–1.14 |
| Systolic BP (mmHg) | 294 | 128.4 ± 15.3 | 92–176 |

**B**

| Framingham risk level: | Male | Female | Total |
|---|---|---|---|
| Low | 50 | 84 | 134 (48%) |
| Intermediate | 56 | 37 | 93 (34%) |
| High | 43 | 7 | 50 (18%) |

**C**

|  | N | Median (IQR) | Range |
|---|---|---|---|
| AHI (events/hr) | 296 | 12.5 (4.4–24.3) | 0.0–103.9 |
| RDI (events/hr) | 296 | 23.9 (14.1–41.1) | 1.1–104.2 |
| AI (events/hr) | 296 | 27.5 (19.2–38.9) | 5.9–100.5 |
| ODI >3% (events/hr) | 296 | 3.8 (0.9–8.8) | 0.0–73 |
| SpO$_2$<90% (%TST) | 296 | 0.4 (0–2.7) | 0.0–98.2 |
| Lowest SpO$_2$ NREM (%) | 296 | 89.0 (86.0–92.0) | 53–96 |
| Lowest SpO$_2$ REM (%) | 288 | 87.0 (81.0–91.0) | 41–97 |
| CCA IMT (mm) | 268 | 0.56 (0.51–0.64) | 0.37–1.01 |
| CFA IMT (mm) | 266 | 0.52 (0.46–0.58) | 0.34–1.03 |

hip ratio), decreased total cholesterol and HDL, and higher SDB metrics (AHI, RDI, ODI >3%, AI) than those not included in the RFM intervention study. Cardiovascular risk levels and IMTs were comparable between the two groups (See S2 and S3 Tables in S1 File).

Significant rank correlations were observed between IMT values and a number of anthropometric, blood chemistry and polysomnography variables (Table 3). Age exhibited the strongest association with CCA IMT (Fig 2A, r = 0.50), while weight was most strongly associated with CFA IMT (Fig 2B, r = 0.32). Fig 2 shows the ln(IMT)-age relationship back-transformed to the original scale of measurement for both CCA (Fig 1A; $R^2$ = 0.25) and CFA (Fig 1B; $R^2$ = 0.07).

**Base models—Main group.** The following variables were entered into a backward stepwise multiple linear regression model of log transformed CCA IMT values: age, BMI, gender, ethnicity, systolic blood pressure, smoking history (combined current and/or past), cholesterol medication, hypertension medication, neck circumference, waist:hip ratio, total cholesterol: HDL ratio, fasting blood sugar level and study group. The final model for ln(CCA IMT) (Table 4A) explained 27% of the variance. Diagnostic plots of the residuals showed good agreement with error assumptions underlying the model. Age was the most significant predictor demonstrating a 9.3% (95%CI 7.0–11.7%, p<0.001) increase in CCA IMT per decade of age. Other significant predictors were systolic BP predicting 0.20% (95%CI 0.04–0.40%, p = 0.016) increase per mmHg and waist:hip ratio predicting 27% (95%CI 1.2–59%, p = 0.040) increase

**Table 2. RFM sub-group demographics and sleep disordered breathing status.** Demographics (A), cardiovascular risk (B), IMT and SDB status (C) of the RF Minimisation Sub-Group, presented as mean ± SD, median (IQR) or frequency, where appropriate, plus range. See text for abbreviations.

**A**

|  | N | Mean ± SD | Range |
|---|---|---|---|
| Age (yrs) | 157 | 57.7 ± 9.8 | 35–79 |
| Height (cm) | 157 | 169.3 ± 10.1 | 144.0–191.5 |
| Weight (kg) | 157 | 92.4 ± 21.4 | 45.8–186.3 |
| BMI (kg/m$^2$) | 157 | 32.2 ± 7.0 | 19.6–64.5 |
| Neck circumference (cm) | 157 | 40.7 ± 4.6 | 31.0–56.5 |
| Waist:hip ratio | 157 | 0.95 ± 0.09 | 0.89–0.96 |
| Systolic BP (mmHg) | 156 | 129.4 ± 14.2 | 92–172 |

**B**

| Framingham risk level: | Male | Female | Total |
|---|---|---|---|
| Low | 33 | 37 | 70 (47%) |
| Intermediate | 34 | 13 | 47 (32%) |
| High | 28 | 4 | 32 (21%) |

**C**

|  | N | Median (IQR) | Range |
|---|---|---|---|
| AHI (events/hr) | 157 | 14.7 (7.2–33.2) | 0.0–103.9 |
| RDI (events/hr) | 157 | 30.7 (17.4–45.3) | 1.4–104.2 |
| AI (events/hr) | 157 | 32.6 (21.3–43.5) | 5.9–100.5 |
| ODI >3% (events/hr) | 157 | 4.4 (1.1–13.7) | 0.0–73.0 |
| SpO$_2$<90% (%TST) | 157 | 0.5 (0–3.4) | 0.0–98.2 |
| Lowest SpO$_2$ NREM (%) | 157 | 90.0 (85.5–92.0) | 53–96 |
| Lowest SpO$_2$ REM (%) | 153 | 87.0 (79.5–91.0) | 41–95 |
| Snores (events/hr) | 155 | 470 (282–652) | 26–1051 |
| CCA IMT (mm) | 157 | 0.56 (0.50–0.64) | 0.37–1.01 |
| CFA IMT (mm) | 155 | 0.52 (0.46–0.60) | 0.34–1.03 |

in CCA IMT per au. Study group was not a significant predictor and was excluded prior to the final model.

The following variables were entered into the backward stepwise multiple linear regression model of log transformed CFA IMT values: age, weight, gender, ethnicity, systolic blood pressure, smoking history, cholesterol medication, hypertension medication, total cholesterol: HDL ratio and study group. The final model (Table 4B) explained 21% of the variance. Diagnostic plots of the residuals showed good agreement with error assumptions underlying the model. Age was the most significant predictor demonstrating a 6.9% (95%CI 4.6–9.3%, p<0.001) increase in CFA IMT per decade of age. Other significant predictors were weight, predicting 0.30% (95%CI 0.10–0.50%, p<0.001) increase in CFA IMT per kg, and total cholesterol:HDL ratio predicting 1.9% (95%CI 0.14–3.7%, p = 0.037) increase per au. Study group was not a significant predictor and was excluded prior to the final model.

**SDB variable models—Main group.** Log transformed SDB variables (see Table 1C plus AHI Category and severe/non-severe OSA) were added individually to the base models for the log transformed IMT values. None of these SDB variables emerged as significant predictors independent of the base models (all p>0.05) for either ln(CCA IMT) (Table 5A) or ln(CFA IMT) (Table 5B). Their addition to the base models resulted in negligible improvement in the proportion of observed variance explained (lnCCA IMT: $\Delta R^2$ from -0.011–0.003; lnCFA IMT: $\Delta R^2$ from -0.063–0.01).

**Table 3. Main group—Spearman rank correlations.** Spearman rank correlation (r) between each variable used in the models and IMT values for CCA and CFA. See text for abbreviations.

| Variable | CCA IMT | | CFA IMT | |
|---|---|---|---|---|
| | r | p-value | r | p-value |
| Age | 0.50 | <0.001[+] | 0.29 | <0.001[+] |
| BMI | 0.04 | 0.50 | 0.26 | <0.001[+] |
| Height | 0.02 | 0.80 | 0.15 | 0.01[+] |
| Weight | 0.02 | 0.70 | 0.32 | <0.001[+] |
| Neck circumference | 0.11 | 0.07 | 0.31 | <0.001[+] |
| WHR | 0.15 | 0.02[+] | 0.23 | <0.001[+] |
| Total cholesterol | -0.09 | 0.10 | -0.03 | 0.70 |
| HDL | 0.01 | 0.80 | -0.09 | 0.10 |
| Total cholesterol:HDL | -0.10 | 0.10 | 0.07 | 0.20 |
| BSL fasting | 0.14 | 0.02[+] | 0.05 | 0.50 |
| Systolic BP | 0.25 | <0.001[+] | 0.21 | 0.001[+] |
| AHI | 0.13 | 0.03[+] | 0.17 | 0.005[+] |
| RDI | 0.15 | 0.02[+] | 0.17 | 0.005[+] |
| AI | 0.15 | 0.01[+] | 0.18 | 0.004[+] |
| ODI >3% | 0.11 | 0.06 | 0.11 | 0.070 |
| $SpO_2$<90% | 0.16 | 0.01[+] | 0.10 | 0.10 |
| Lowest $SpO_2$ NREM (%) | -0.183 | 0.003[+] | -0.153 | 0.013[+] |
| Lowest $SpO_2$REM (%) | -0.133 | 0.032[+] | -0.027 | 0.67 |

[+] Indicates significant (p<0.05).

**Base models—RFM sub-group.** The study group variable ('included' in RFM Sub-Group versus 'not included') was not significant for any of the cross-sectional base models. Consequently, the base models for the RFM Sub-Group were constructed using the same variables as emerged in the Main Group base model results shown above.

For ln(CCA IMT), the base model explained 27% of the variance. Age and systolic BP were significant predictors with age demonstrating a 9.2% (95%CI 6.2–12.2%, p<0.001) increase in CCA IMT per decade of age and systolic BP predicting 0.20% (95%CI 0.00–0.40%, p = 0.033) increase in CCA IMT per mmHg (See S4A Table in S1 File). For ln(CFA IMT), the base model explained 19% of the variance. Age and weight were significant predictors with age demonstrating a 7.5% (95%CI 4.5–10.8%, p<0.001) increase in CFA IMT per decade of age and weight predicting 0.30% (95%CI 0.10–0.50%, p<0.001) increase in CFA IMT per kg (See S4B Table in S1 File).

**SDB variable models—RFM sub-group.** None of the interactions between study group and each SDB variable [SDB variable x study group] adjusted for the base model interactions had a significant effect on log transformed CCA IMT values in the Main Group data. Consequently no SDB variable testing was undertaken in the RFM Sub-Group (See S5A Table in S1 File).

For log transformed CFA IMT values, the following significant interactions were detected in the Main Group after adjusting for the base model: ln(AHI) x study group (p = 0.01), ln(RDI) x study group (p = 0.03), ln(ODI >3%) x study group (p = 0.02), ln($SpO_2$<90%) x study group (p = 0.01) and ln (Lowest $SpO_2$ REM)*study group (p = 0.04). This indicated that the relationship between ln(CFA IMT) and these SDB variables required separate investigation in the RFM Sub-Group (See S5B Table in S1 File). When added individually to the base model in the RFM Sub-Group, only ln(AHI) ln(RDI) and ln(AI) had a significant effect on ln(CFA

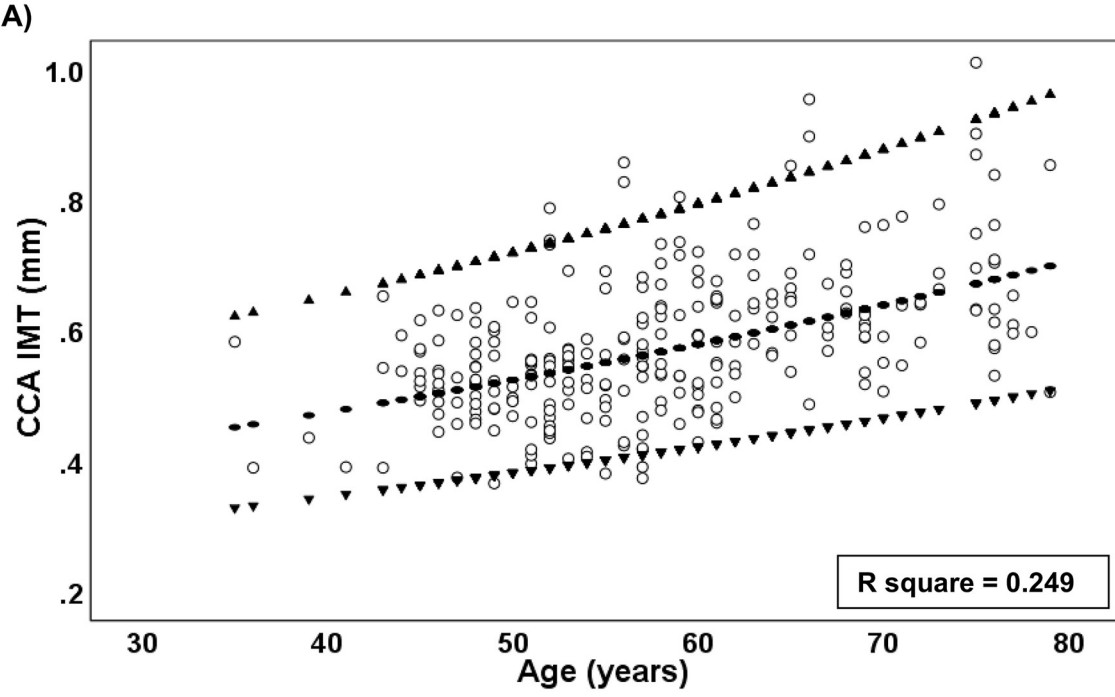

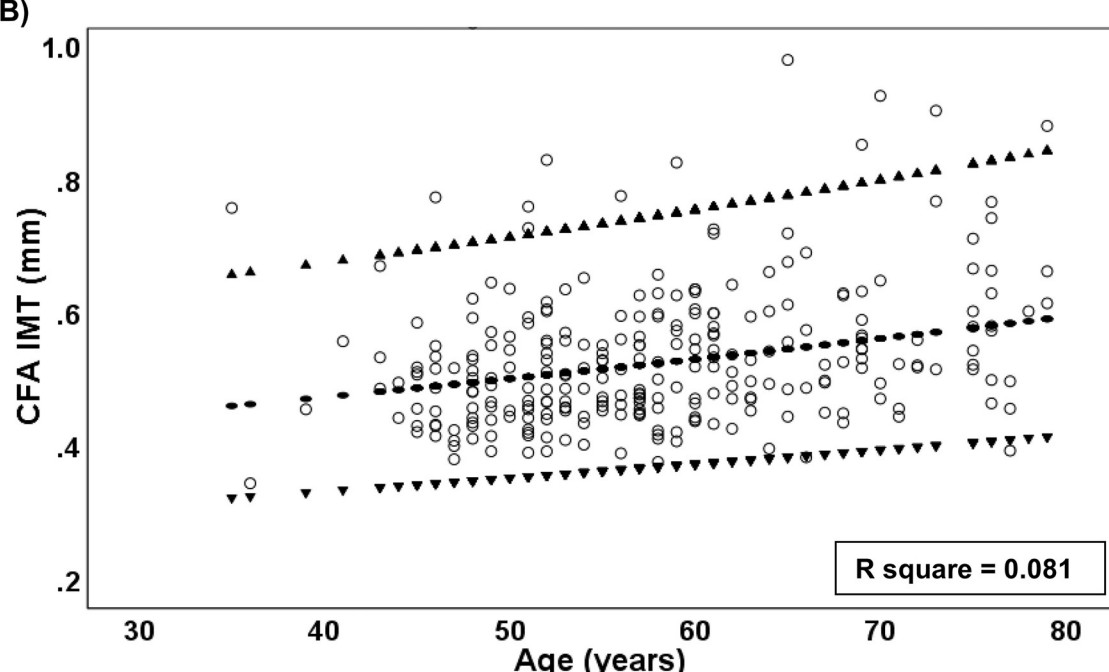

**Fig 2. Main group IMT-age relationships.** Scatter plots of IMT versus age for CCA (A) and CFA (B). The back-transformed predicted values and 95% confidence bands from the linear regression of ln(IMT) on age are shown. $R^2$ for the relationship is higher for the CCA than for the CFA, but both have wide predictive bands for IMT indicating significant non-age-related variability.

**Table 4. Main group IMT base models.** The base models for log transformed CCA IMT (A) and CFA IMT (B) showing the independent, non-SDB predictor variables for the Main Group.

| Variable | B | S.E. | p-value |
|---|---|---|---|
| **A ln(CCA IMT)** | | | |
| (Constant) | -1.508 | 0.134 | <0.001 |
| Age (per decade) | 0.089 | 0.011 | <0.001 |
| Waist:hip ratio (au) | 0.237 | 0.115 | 0.040 |
| Systolic BP (mmHg) | 0.002 | 0.001 | 0.016 |
| **$R^2$ = 0.270** | | | |
| **B ln(CFA IMT)** | | | |
| (Constant) | -1.378 | 0.093 | <0.001 |
| Age (per decade) | 0.067 | 0.011 | <0.001 |
| Weight (kg) | 0.003 | 0.001 | <0.001 |
| Total cholesterol:HDL ratio | 0.019 | 0.009 | 0.037 |
| $R^2$ = 0.206 | | | |

B = unstandardized beta coefficient; S.E. = standard error of B.

IMT) (See S6 Table in S1 File). For each 2-fold increase in AHI there was a predicted 2.4% (95%CI 0.5%-4.4%, p = 0.01) increase in CFA IMT, and for each 2-fold increase in RDI there was a predicted 4.0% (95%CI 1.0%-7.2%, p = 0.009) increase in CFA IMT after adjusting for the base model.

**Table 5. Main group SDB variable models for IMT.** Results of adding each SDB variable individually to the base model for log transformed IMT values, i.e. base model + SDB variable, for the CCA IMT (A) and CFA IMT (B) in the Main Group. See text for abbreviations.

| Variable | B* | S.E.* | p-value* | $\Delta R^2$ |
|---|---|---|---|---|
| **A ln(CCA IMT)** | | | | |
| ln(AHI) | -0.004 | 0.009 | 0.70 | 0.001 |
| ln(RDI) | <0.001 | 0.013 | >0.90 | 0.0 |
| ln(AI) | -0.003 | 0.02 | 0.90 | 0.0 |
| ln(ODI >3%) | 0.001 | 0.009 | 0.90 | 0.0 |
| ln(SpO$_2$<90%) | 0.002 | 0.01 | 0.90 | 0.0 |
| ln (Lowest SpO$_2$ NREM) (%) | 0.008 | 0.127 | 0.95 | 0.0 |
| ln (Lowest SpO$_2$ REM) (%) | -0.027 | 0.078 | 0.73 | -0.011 |
| AHI category | - | - | 0.80 | 0.003 |
| AHI >30 events/hr | -0.013 | 0.025 | 0.6 | 0.001 |
| **B ln(CFA IMT)** | | | | |
| ln(AHI) | 0.012 | 0.01 | 0.2 | 0.005 |
| ln(RDI) | 0.024 | 0.014 | 0.09 | 0.009 |
| ln(AI) | 0.038 | 0.021 | 0.07 | 0.01 |
| ln(ODI >3%) | 0.002 | 0.01 | 0.8 | 0.0 |
| ln(SpO$_2$<90%) | -0.007 | 0.01 | 0.5 | 0.001 |
| ln (Lowest SpO$_2$ NREM) (%) | -0.059 | 0.147 | 0.689 | -0.063 |
| ln (Lowest SpO$_2$ REM) (%) | -0.029 | 0.087 | 0.736 | -0.056 |
| AHI category | - | - | 0.5 | 0.007 |
| AHI >30 events/hr | 0.038 | 0.026 | 0.1 | 0.007 |

B* = unstandardized beta coefficient adjusted for base model; S.E.* = standard error of B*; $\Delta R^2$ is the incremental change in $R^2$ after addition of the SDB variable to the Base Model calculated as (SDB model $R^2$- Base Model $R^2$).

### RFM intervention study

For the RFM sub group (n = 137) the mean AHI value after titration was 1.7 ± 2.1 events/hr.

Participants used CPAP for 5.0 ± 2.2 hrs/night (mean ± SD), with 74% of the group having CPAP compliance of ≥ 4 hrs/night across the 12-month intervention.

**CCA IMT.** After 12 months on study, there was a non-significant overall 0.8% (95%CI -1.4%, 3.1%, p = 0.476) increase in CCA IMT within participants from baseline. The size of the change was associated with weight change (r = 0.21, p = 0.016): CCA IMT decreased by 3.8% (95%CI 0.1–7.2%, p = 0.048) in those who lost weight and increased by 3.3% (95%CI 0.5–6.1%, p = 0.021) in those who did not. Linear mixed effects models of ln(CCA IMT) demonstrated this effect was essentially unchanged after correcting for the fixed baseline effects (age, waist: hip ratio and systolic BP) and that there was no evidence of a CPAP effect on the within partipant change in ln(CCA IMT) (interaction p = 0.512 between Visit and CPAP use); Fig 3A.

**CFA IMT.** After 12 months on study, there was a significant 3.5% (95%CI 0.8, 6.2%, p = 0.013) overall reduction in CFA IMT within participants from baseline. The linear mixed effects model of ln(CFA IMT) adjusted for the fixed baseline effects (age, weight and total cholesterol:HDL ratio) demonstrated a 2.9% (95%CI 0.3, 5.5%, p = 0.028) adjusted reduction in CFA IMT. There was no evidence of a CPAP effect on the within partipant change in ln(CFA IMT) (interaction p = 0.464 between Visit and CPAP use). There was, however, significant rank correlation between Δln(CFA IMT) and Δweight (r = 0.25; p = 0.004). When weight loss status and its 2 and 3-way interactions with Visit and CPAP use were added as fixed effects to the LME model for ln(CFA IMT), a significant 3-way interaction was detected (p = 0.020) indicating that the change in ln(CFA IMT) depended on the joint effects of weight loss status and CPAP use. In particular, there was a 12.9% (95%CI 6.8, 18.7%, p = 0.001) reduction in CFA IMT from baseline in those who both lost weight and used CPAP > = 4hours/day, but no significant changes in CFA IMT were observed in those who gained/maintained weight or used CPAP <4hours/day (Fig 3B).

## Discussion

In a non-diabetic, sleep clinic cohort, aged >35 yrs, carotid and femoral artery IMT values were positively associated with the participant's age. Statistical models incorporating a range of non-sleep and SDB variables explained at most 27% of the variance in CCA IMT and 21% of the variance in CFA IMT, leaving the majority of the between individual differences unexplained. Non-sleep related predictive factors for CCA and CFA IMT were broadly in alignment with those identified by large population studies [27–31]. Smoking history, however, did not emerge as a signfiicant predictive factor for CCA and CFA IMT, this is probably because there were very few current (most at risk) smokers (n = 11) in the cohort.

We did not identify any impacts of SDB severity on CCA IMT values. However, in a subgroup with a wide range of SDB severity, RDI and AHI emerged as predictive variables for CFA IMT: a 2-fold increase in RDI predicting a 4.0% increase in CFA IMT, and a 2-fold increase in AHI predicting a 2.4% increase.

When sub-group individuals were stabilised on medical therapy, monitored and given CPAP for 12 months, CCA IMT increased within participant by a nonsignificant 0.8% in accordance with age. In contrast, CFA IMT decreased within participant by on average 3.5%. This reduction was related to both weight loss and CPAP use with those who lost weight and also used CPAP > = 4hrs/day demonstrating a reduction in CFA IMT from baseline.

**A)**

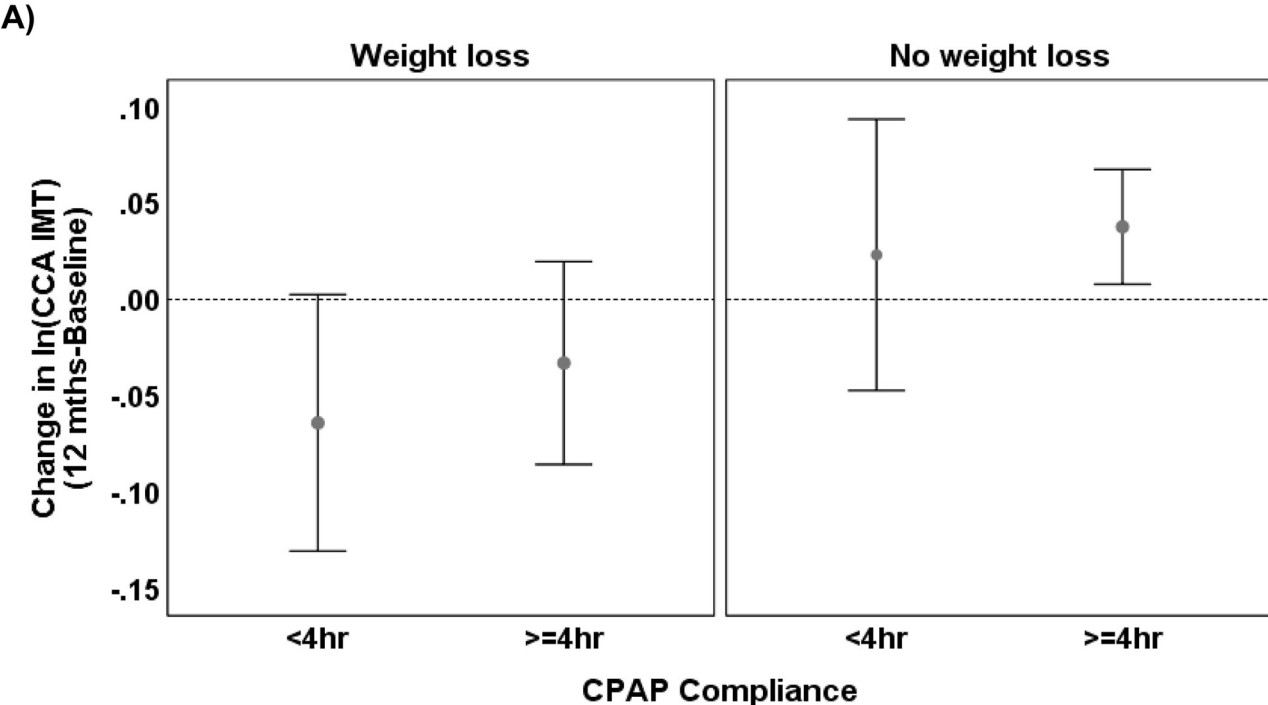

**B)**

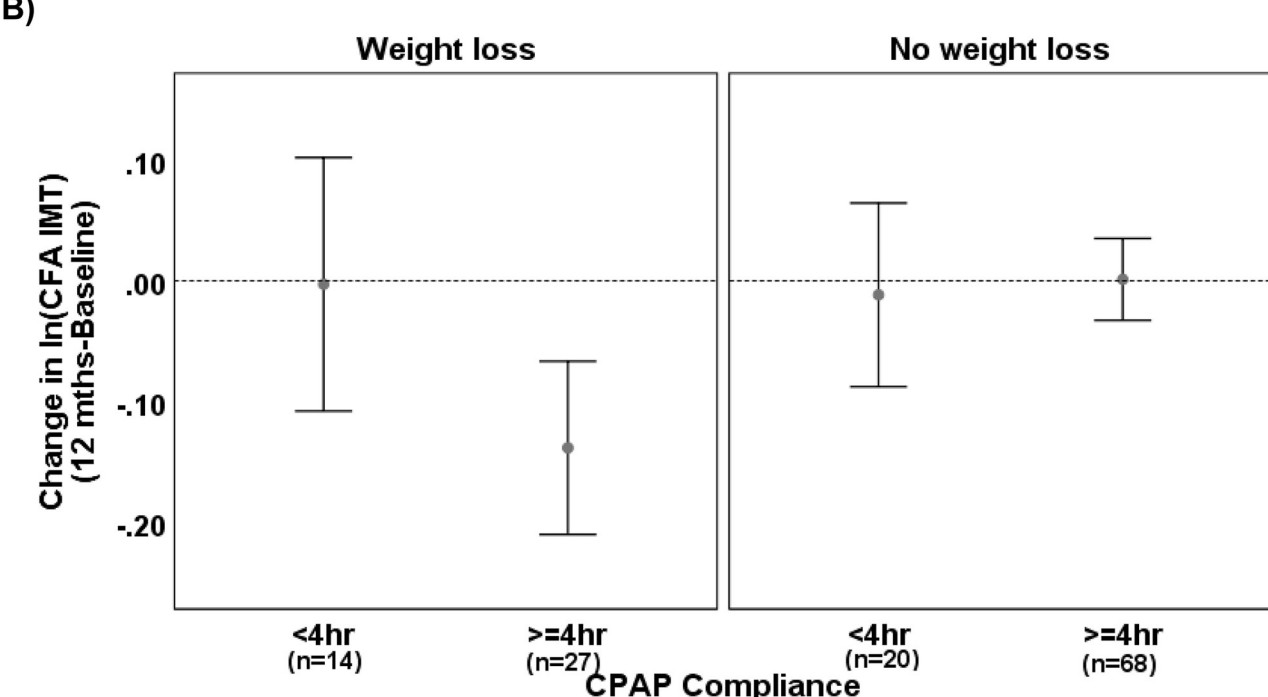

**Fig 3. RFM sub-group—Effect of CPAP use on ln(CCA IMT) and on ln(CFA IMT) by weight loss status.** Mean with 95%CIs showing the within participant change in ln(CCA IMT) and in ln(CFA IMT) in the RFM Sub-Group during the 12-month RFM intervention by weight loss status and CPAP use. The ln(CFA IMT) depended on both weight loss and CPAP use (p = 0.020) with those who lost weight and also used CPAP > = 4hrs/day demonstrating a 12.9% (95%CI 6.8, 18.7%, p = 0.001) reduction in CFA IMT from baseline compared to no significant change in those who gained/maintained weight or used CPAP <4hours/day.

## Study cohort characteristics

In interpreting the results of the present study, it is important to understand the characteristics of the study cohort. This is not a randomised community-recruited population-based study. Rather, we targeted a specific Main Group (i.e. patients referred to a sleep clinic for assessment of SDB) and then screened for and eliminated diabetes as a RF. Consequently, our study group is also not a randomised sleep clinic population sample. The aim was to test for associations between specific SDB metrics and specific macro-vascular characteristics (IMT) in an environment where the influence of known cardio-vascular disease RFs was minimised. The rationale for this approach assumed that SDB influences were likely to be small compared with well-known systemic RFs, such as diabetes and smoking, and therefore, would be more likely to be detected in the absence of these major confounders. A number of previously published studies that have attempted to quantify associations between carotid IMT and SDB have highlighted this limitation [11, 32, 33].

In the Main Group not only was diabetes eliminated but ~40% of participants were on current treatment for hypertension and/or hypercholesterolaemia. Blood lipid and sugar levels were centred on normal levels for the group, but with a range of individual values (See S1C Table in S1 File). Framingham Cardiovascular Risk levels were relatively evenly distributed between low and intermediate/high, tended to be higher in males, and were similar to those reported for population studies across this age range [34, 35]. Sleep disordered breathing varied across the severity spectrum but was skewed to lower values, with only ~40% being determined to have moderate/severe OSA (AHI ≥ 15 events/hr).

It is also important to understand the nature of the RFM Sub-Group, in which an association between SDB and femoral artery IMT was detected. The RFM Sub-Group had a similar Framingham Cardiovascular Risk profile to the Main Group, but a greater proportion were male, and so had corresponding changes in body size measures (taller, weighed more, larger neck circumference). Overall SDB was more severe in the RFM Sub-Group with ~51% considered to have moderate/severe OSA. Thus, the OSA distribution of the Main Group skewed towards normal-mild, while the RFM Sub-Group, by design, had a more even distribution.

## CCA IMT

CCA IMT is an established marker for cardiovascular and cerebrovascular risk [9, 10]. It has strong associations with established cardiovascular RFs including age, gender, BMI, lipid profile, diabetes, smoking and ethnicity [28, 29, 36], and these RFs have been variably used in regression modelling to predict CCA IMT in healthy and cardiovascular disease populations [27, 28]. OSA has been found to be independently associated with CCA IMT [32, 37–39] in sleep clinic cohorts, with a recent meta-analysis finding a moderate correlation (r = 0.389) between AHI and CCA IMT [11]. However, differences in the degree of heterogeneity, small sample sizes, as well as differences in screening procedures and analysis techniques, create a degree of uncertainty as to the generalisability of these results.

Values for CCA IMT in the present study were positively associated with age and fell in the reported population normal range for age [27, 40]. Consequently, our non-diabetic sleep clinic-recruited cohort values for CCA IMT do not appear to vary systematically from what would be expected for a wider community age-matched group. Of the range of potential predictive variables tested in our multiple linear regression base models, age was the strongest independent predictor for ln(CCA IMT) along with minor contributions from systolic blood pressure and a body size measure (waist-hip ratio). This model, however, only explained ~27% of the total variance, leaving the majority of the difference between individual subject values unexplained.

When SDB variables were added to our base models there was either no or minimal improvement in the amount of variance explained and no SDB variable emerged as a significant independent predictor. Consequently, in this group, with a range of cardiovascular risk and where the distribution of SDB severity was skewed to lower values, we could not detect a role in predicting CCA IMT values for any SDB severity metric. This result also applied to the RFM Sub-Group with its greater relative proportion of more severe SDB. Interestingly, CCA IMT demonstrated a nonsignificant 0.8% overall mean increase within the Sub Group during the 12 month intervention in keeping with the predicted 0.9% age effect detected in the Main Group base model (see Table 4A). Linear mixed effects models of ln(CCA IMT) adjusted for the base model variables detected a significant difference between the 12 month 3.8% mean reduction in CCA IMT observed in those who lost weight compared to the 3.3% mean increase in those who gained weight. No significant association with CPAP use was seen.

The rank correlations between CCA IMT and SDB variables observed in the cross-sectional study are consistent with reports in the literature, but our rank correlation with AHI is lower than that reported elsewhere [11, 32]. We were unable to detect an independent association between SDB and ln(CCA IMT) using multiple regression analysis. This finding contrasts with reports in the literature of a significant independent association between CCA IMT and AHI values [32, 38]. There were, however, several key differences between these studies and ours. Suzuki et al. [38] studied a sleep clinic cohort, without excluding diabetes, with more severe SDB and higher IMT values, while Drager et al. [32] studied a small, highly selected group, excluding most other RFs, including age. The low IMT values in the present study–CCA IMT median (IQR): 0.56 (0.51–0.64) mm, i.e. very few IMT's in the abnormal range–may have contributed to the weaker association seen with AHI compared to others. This may in turn have influenced our ability to find an independent association in multivariable models. Our results, however, were broadly in alignment with Kim et al. [20] who studied a similarly sized group with medically controlled RFs. Their findings of no significant difference in CCA IMT between participants with OSA and controls and no effect of CPAP on CCA IMT after 4 months are consistent with the present study. By excluding diabetes and controlling other RFs we may have limited the possibility for interactions between SDB and these other RFs that have previously been reported to have additive effects on IMT [33].

## CFA IMT

CFA IMT has been studied much less than CCA IMT, but similar associations with cardiovascular RFs have been found [13]. Indeed, emerging evidence suggests that CFA IMT may actually be a stronger predictor of coronary artery disease [15] and a more sensitive measure of overall cardiovascular risk [12]. To date, no other studies have examined relationships between SDB and CFA IMT.

Values for CFA IMT in the present study were positively associated with age, but not as strongly as for CCA IMT, and were broadly aligned with the limited population normal range for age available in the literature [41, 42] and did not differ between the Main and RFM Sub-Groups (Table 4). In contrast to CCA IMT, CFA IMT demonstrated rank correlations of approximately 0.3 with body size metrics, especially weight and neck circumference, and weaker positive associations with systolic blood pressure and all SDB variables, except those for hypoxia (Table 2). These findings suggest that age and body size are the main univariable associative factors for CFA IMT operating in this data set, and, in comparison, the rank correlations with SDB variables are weaker and less certain.

In the Main Group multiple linear regression analysis, age and weight were identified as the strongest independent predictors for ln(CFA IMT), with total cholesterol:HDL ratio also

emerging as significant predictor. However, the model itself only explained 21% of the total variance leaving the majority of the variance unexplained. When SDB variables were added to the base model there was either no or minimal improvement in the amount of variance explained and no SDB variable emerged as a significant independent predictor. Consequently, in this group, with a range of cardiovascular risk and where the distribution of SDB severity was skewed to lower values, we could not detect a role for any SDB severity metric in predicting CFA IMT values.

There were, however, significant interactions between several SDB variables and study group, i.e. the effect of SDB variables on ln(CFA IMT) adjusted for the base model differed between those included in the RFM Sub-Group and those not included. Both ln(AHI) and ln (RDI) emerged as significant independent predictors of ln(CFA IMT), explaining an additional 3–4% of the variance when added to the base model in the RFM Sub-Group. This differs from the Main Group outcome and suggests a potential small role for SDB severity as a predictor of CFA IMT in a Sub-Group enriched with more severe SDB.

There are no previous studies examining CFA IMT in OSA. There are, however, a number of publications that support differing biology and predictors [12, 13, 15] for CFA versus CCA IMT. In the present study different predictors were identified in the Main Group and RFM Sub-Group. Given that the RFM Sub-Group contained a greater proportion of males with more severe SDB, one potential interpretation is that this finding reflects more systemic impacts of SDB operating in the RFM Sub-Group. We speculate that the femoral vascular bed may be more sensitive than the carotid vasculature to the cardiovascular stressors seen during sleeping with OSA (intermittent hypoxia and increased nocturnal sympathetic activity).

CFA IMT decreased significantly within participants by 3.5% (95%CI 0.8, 6.2%, p = 0.013) with 12 months of RFM intervention. Considering that the CFA IMT-age relationship for this group predicted a 0.7% increase, the observed decrease is additional to this expected increase. Linear mixed effects modelling of ln(CFA IMT) adjusted for the base model variables detected a significant association between the 12 month change observed in ln(CFA IMT) and the joint effects of CPAP use and within patient weight change: those who lost weight and also used CPAP > = 4hrs/day reduced CFA IMT from baseline compared to no significant change in those who gained weight or used CPAP <4hours/day. These findings suggest that treatment of SDB with CPAP, when combined with weight loss and medical stabilisation of vascular RF, may help contribute to reducing CFA IMT thickening, at least over the first year post diagnosis. Further studies will be require to determine if this effect is maintained over longer time frames.

## Strengths & limitations

This study has a number of strengths and limitations. The small study size limits statistical power to detect relatively small changes in IMT against a background of remaining more influential RFs. The established non-SDB RFs were only able to explain 20–27% of the variance in IMT for the group. This is comparable with the results of larger population studies that have found similar RFs explaining 29% of the variance in CCA IMT [28, 43]. The fact that we were only able to detect small independent associations between SDB variables and CFA IMT in our OSA-enriched RFM Sub-Group suggests that detecting such associations in an overall sleep clinic population would require larger numbers. However, the Main Group size compares favourably with other studies examining associations between SDB and IMT and the RFM Sub-Group is one of the largest studies, with a long follow-up period, examining the association between changes in IMT and RFM intervention (including CPAP treatment).

The composition of the study group: 1) eliminated a major RF, diabetes; 2) controlled for some, i.e. treatment of hypertension and hypercholesterolaemia; and 3) allowed a range of other RFs, e.g. age and body size. This was an attempt to avoid trying to find a potentially small SDB effect in the presence of more influential confounders typical of larger groups, while simultaneously avoiding very small study groups produced by eliminating all known RFs. Our efforts may, however, have succeeded in dampening interaction effects between classic RFs and SDB by monitoring and controlling them carefully. This may have resulted in IMT values for the study group that essentially fell within the normal range predicted from larger population studies. This increased the difficulty to detecting independent associations with SDB and was also an important limitation for the RF Minimisation Intervention. A "floor effect" may have been in play, whereby IMT levels could not be further reduced. Nevertheless, we were able to detect a significant overall decrease in CFA IMT across the intervention period, while CCA IMT tended to increase in line with the 1-year increase in age.

Examination of the femoral artery was an important strength of the study. The stronger associations found between SDB and CFA versus CCA IMT suggest that the CFA IMT may be a better marker of cardiovascular disease than the CCA IMT. Our multivariable models support a role for SDB as a small, but significant, independent predictor of CFA IMT. Given the emerging evidence regarding the differences in biology between the carotid and femoral arteries, the CFA IMT may also be a useful marker of SDB-associated cardiovascular risk.

## Summary

Overall findings from the cross-sectional study identified classic vascular RFs for IMT. Although we found univariable associations for metrics of SDB severity, we did not identify any independent associations in the Main Group. In the more severely SDB affected RF Minimisation Sub-Group, both AHI and RDI emerged as independent predictors of CFA IMT. A 12-month RF Minimisation Intervention reduced CFA IMT by 12.6% in those who both lost weight and used CPAP $>= 4$hours/day.

## Conclusion

In this study of non-diabetic patients presenting to a sleep clinic, total variance in carotid and femoral artery IMT was only modestly explained by a combination of anthropometric characteristics (predominantly age) and known clinical RFs. There was no detectable contribution from SDB metrics, except for a small contribution to CFA IMT variance attributable to measures of SDB severity (AHI/RDI). Intervening with 12 months of RF minimisation including CPAP did not alter CCA IMT but reduced CFA IMT specifically in those who both lost weight and used CPAP $>= 4$hours/day.

We conclude that once classical vascular disease RFs are excluded or controlled, SDB may play a minor role in increasing risk for femoral, but not carotid, artery IMT values in sleep clinic patients. This effect, however, is minimal when compared with the influence of known cardiovascular RFs and is even less substantial when compared with the levels of unexplained variance revealed by our modelling. However, this is the first study to suggest a potential linkage between femoral artery IMT levels and SDB severity.

## Supporting information

**S1 File.**
(DOCX)

**S1 Protocol.**
(PDF)

**S2 Protocol.**
(PDF)

**S1 Checklist. TREND statement checklist.**
(PDF)

**S1 Dataset. IMT data files.**
(ZIP)

**S2 Dataset. SPSS data files.**
(ZIP)

**S1 Codes. SPSS syntax codes.**
(ZIP)

## Acknowledgments

We thank Tracey Burns, Anne Drury, Warde Elias, Allison Mitchell, Ayey Susan Madut, Victoria Sissanes, Meredith Wickens, and Heather Wood for assistance with the study.

## Author Contributions

**Conceptualization:** George Larcos, Kristina Kairaitis, Terence Amis, John Wheatley.

**Data curation:** Christopher Lambeth, Rita Perri, Sharon Lee, Manisha Verma, Nicole Campbell-Rogers.

**Formal analysis:** Christopher Lambeth, Rita Perri, Karen Byth, Terence Amis.

**Funding acquisition:** Terence Amis, John Wheatley.

**Investigation:** Christopher Lambeth, Rita Perri, Sharon Lee, Manisha Verma, Nicole Campbell-Rogers, George Larcos, Kristina Kairaitis.

**Methodology:** George Larcos, Kristina Kairaitis, Terence Amis, John Wheatley.

**Project administration:** Sharon Lee.

**Resources:** Kristina Kairaitis, John Wheatley.

**Supervision:** Sharon Lee, Kristina Kairaitis, Terence Amis, John Wheatley.

**Visualization:** Christopher Lambeth.

**Writing – original draft:** Christopher Lambeth, Terence Amis.

**Writing – review & editing:** Christopher Lambeth, Karen Byth, Kristina Kairaitis, Terence Amis, John Wheatley.

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
