## [Decision Letter · Decision Letter 0]

15 Dec 2020

PONE-D-20-02731

Predictors for carotid and femoral artery intima-media thickness in a non-diabetic sleep clinic cohort

PLOS ONE

Dear Dr. Amis,

Thank you for submitting your manuscript to PLOS ONE. After careful consideration, we feel that it has merit but does not fully meet PLOS ONE’s publication criteria as it currently stands. Therefore, we invite you to submit a revised version of the manuscript that addresses the points raised during the review process.

I would like to sincerely apologise for the delay you have incurred with your submission. It has been exceptionally difficult to secure reviewers to evaluate your study. We have now received three completed reviews; their comments are available below. Reviewer#1 and #3 have raised concerns about the statistical analysis in this study that need to be addressed in a revision.

Please revise the manuscript to address all the reviewer's comments in a point-by-point response in order to ensure it is meeting the journal's publication criteria. Please note that the revised manuscript will need to undergo further review, we thus cannot at this point anticipate the outcome of the evaluation process.

We look forward to receiving your revised manuscript.

Kind regards,

Miquel Vall-llosera Camps

Senior Editor

PLOS ONE

Journal Requirements:

**Comments to the Author**

1. Is the manuscript technically sound, and do the data support the conclusions?

Reviewer #1: Partly

Reviewer #2: Yes

Reviewer #3: Partly

2. Has the statistical analysis been performed appropriately and rigorously? 

Reviewer #1: No

Reviewer #2: Yes

Reviewer #3: Yes

3. Have the authors made all data underlying the findings in their manuscript fully available?

Reviewer #1: No

Reviewer #2: Yes

Reviewer #3: Yes

4. Is the manuscript presented in an intelligible fashion and written in standard English?

Reviewer #1: Yes

Reviewer #2: Yes

Reviewer #3: Yes

5. Review Comments to the Author

Reviewer #1: The manuscript addresses an interesting topic. The collected data contain several features to be analysed. The research questions are appropriate and the employed statistical methods generally sound. Some comments follow.

1. Data are not fully available without restrictions. This is not in line with the journal's guidelines. More importantly, the reviewer does not have the chance to fully check the correctness of the methods. Please, upload the data and the code used to obtain the results.

2. Statistical tests and models are applied without a full investigation of the assumptions which must be fulfilled to ensure the reliability of the data. This is of course true for the test on linear correlations, but even more important when the regression model is employed. The authors must provide evidence that the Gauss-Markov assumptions are fulfilled. The analysis of residuals must be included.

3. The main outcome is skewed. A log transformation is applied. However, this is not justified and lead to a different empirical model, where the outcome has a different shape (i.e. a different variability). I am wondering why a regression model with skewed errors is overlooked. I see that "usually people act this way", but this does not mean that it is correct.

4. The performed longitudinal analysis is quite obscure to me. Do you have missing values? How do you account for repeated measurements and, accordingly, to dependence across obsevartions belonging to the same unit? You mention the linear mixed model, which is sound. Do you assumed Gaussian random effects? Is it a reasonable assumption for the data at hand? Wha about random coefficients? Again, please, check model's assumptions.

a. The R^2 are accompanied by a p-value. What does it refer to? Are you really comparing the intercept-only model with a model where the covariates are significant, and does obviuously improve the fit? In general, model fitting is rather poor, so there is a lot of unexplained heterogeneity.

Reviewer #2: This is an interesting article investigating the relationship between IMT of CCA/CFA and polysomnographic parameters. The results showed no significant association between severity of sleep apnea and carotid IMT. However, small association was found between femoral IMT and AHI/RDI. Otherwise, CPAP therapy for 12 months did not alter carotid IMT but reduced femoral IMT that was correlated with weight change instead of CPAP use.

Major comments

1.Some important variables such as mini-O2 and snoring can be associated with carotid IMT and need to be included for analyses.

2.The change of IMT in one year period could be very tiny. Thus persistent follow-up is necessary to clarify the long-term effect. Regarding CPAP outcome in terms of IMT, no progression is another explanation in comparison to progression of IMT in non-treatment group patients. This can be put into discussion or do subgroup analysis to clarify the CPAP effect.

3.The clinical meaning and significance of femoral IMT in contribution to OSA need discussion.

Reviewer #3: Review of the manuscript “Predictors for carotid and femoral artery intima-media thickness in a non diabetic sleep clinic cohort”

The study is interesting, however, there are some main concerns to be evaluated. Changing several variables in the 12-month sub-study can leave the analyzes confusing. BMI and OSA severity are collinear variables. Even when conducting collinearity tests, with a relatively small number of patients, we may have erroneous conclusions. Another fundamental point is the inclusion of smokers in this study. Smoking increases cardiovascular risk by 20 x, so it is difficult to analyze risk factors of such different importance in a study of approx. 200 participants. Please insert more information about smoking in the text

Authors must inform cpap adherence data, which is essential for data interpretation. The SAVE Study was cited, but the main criticism of the study is precisely the low adherence to cpap. Therefore, a reduction in cardiovascular risk with cpap cannot be assessed without this analysis. The adherence group (> 4 h of daily use) vs the non-adherence group (<4 h of daily use) should be evaluated

Minor comments:

Describe whether fixed or automatic cpap was used. Describe the AIH after titration

6. PLOS authors have the option to publish the peer review history of their article (what does this mean?). If published, this will include your full peer review and any attached files.

Reviewer #1: No

Reviewer #2: No

Reviewer #3: **Yes: **Rodrigo Pinto Pedrosa

---

## [Author Response · Author response to Decision Letter 0]

20 Apr 2021

Response to reviewers (as per uploaded response file)

RESPONSE TO REVIEWERS

Reviewer #1: “The manuscript addresses an interesting topic. The collected data contain several features to be analysed. The research questions are appropriate and the employed statistical methods generally sound. Some comments follow.”

1.1 Reviewer Question: “Data are not fully available without restrictions. This is not in line with the journal's guidelines. More importantly, the reviewer does not have the chance to fully check the correctness of the methods. Please, upload the data and the code used to obtain the results”.

Response: We have uploaded our data set as “Supplemental Material”.

1.2. Reviewer Question: “Statistical tests and models are applied without a full investigation of the assumptions which must be fulfilled to ensure the reliability of the data. This is of course true for the test on linear correlations, but even more important when the regression model is employed. The authors must provide evidence that the Gauss-Markov assumptions are fulfilled. The analysis of residuals must be included”.

Response: More details have been added to the Methods and Statistical Analysis section (see Revised Manuscript Pages 11-13, Lines 206-266). In particular it is noted there (see Revised Manuscript Page 11, Lines 208-9) and in the Results sections (see Revised Manuscript Table 3) that all reported correlations between continuous variables are Spearman rank correlations (not Pearson correlations which are sensitive to departures from normality). It is also now noted that (1) ‘Diagnostic residual plots (including residuals versus fitted values and Normal probability plots of the residuals) were used to check the adequacy of the fitted models’ in the multiple linear regression analyses and (2) for the linear mixed effects models ‘Diagnostic plots were used to assess the adequacy of the fitted models. They included scatter plots of standardized residuals by fitted values and Normal probablity plots of residuals and of estimated random effects to check the normality assumption for the within-subject errors and for the random effects.’

1.3. Reviewer Question: “The main outcome is skewed. A log transformation is applied. However, this is not justified and lead to a different empirical model, where the outcome has a different shape (i.e. a different variability). I am wondering why a regression model with skewed errors is overlooked. I see that "usually people act this way", but this does not mean that it is correct”.

Response: The Statistical Analysis section (see Revised Manuscript Page 11, Lines 216-218) now notes that the IMT and continuous SDB variables were skewed and were log transformed to approximate Normality and to stabilise the variance prior to analysis. Model parameter estimates for these variables were back transformed and relationships are now reported in terms of relative (rather than absolute) change on the original scale of measurement. The Figure 2 scatterplots of CCA IMT and CFA IMT versus age (see Revised Manuscript Figure 2) now show correct predicted values and 95% confidence bands for the IMT-age relationship based on the model for ln (IMT).

The linear regression models of ln (IMT) more closely adhered to underlying statistical assumptions when diagnostic plots of residuals for models of ln (IMT) were compared with those for IMT (see plots below). We preferred the classical linear regression models for ln (IMT) and believe that these models based on relative change relationships are biologically plausible and easily interpreted by clinicians. Many diagnostic plots were considered and, therefore, we have not included them in the manuscript.

Base models – some diagnostic residual plots for comparison 

 Ln(CCA IMT) (base model Rsq is 0.270) CCA IMT (base model Rsq is 0.264)

 (a) Normal probability plots of residuals

 (FIGURE IN ATTACHED RESPONSE FILE)

 (b) Residuals versus Fitted values

 (FIGURE IN ATTACHED RESPONSE FILE)

Ln (CFA IMT) (base model Rsq =0.206) CFA IMT (model Rsq = 0.191)

 (a) Normal probability plots of residuals

 (FIGURE IN ATTACHED RESPONSE FILE)

 (b) Residuals versus Fitted values

 (FIGURE IN ATTACHED RESPONSE FILE)

1.4. Reviewer Question: “The performed longitudinal analysis is quite obscure to me. Do you have missing values? How do you account for repeated measurements and, accordingly, to dependence across observations belonging to the same unit? You mention the linear mixed model, which is sound. Do you assume Gaussian random effects? Is it a reasonable assumption for the data at hand? What about random coefficients? Again, please, check model's assumptions.”

Response: Data for the longitudinal analysis were collected at only 2 time points, baseline (Visit 1), and 12 months (Visit 2). Patients in the RFM Sub-Group were included in the linear mixed effects model analyses only if they had CCA IMT values recorded at both times (n=137). Two of these patients did not have CFA IMT recorded at 12 months. The only other variables considered in the LME models which had missing data were Average CPAP use during the 12 month intervention (missing for 6/137 patients) and weight (not recorded at 12 months for 3/137 patients, one of whom was also missing CPAP).

As part of the response to this particular reviewer question our statistician has now revisted and extended the LME model analysis using IBM SPSS software (see also Reviewer Question 3.3 re CPAP compliance analysis). In these models of the log transformed IMT values, subject was used as the group identifier. The ‘Visit’ (1=baseline, 2=12 months) variable was fitted as both a random effect with unstructured covariance matrix and as a fixed effect. The covariates from the relevant base model were added along with average CPAP use and its interaction with Visit as fixed effects. A significant (CPAPxVisit) interaction term was interpreted as evidence that CPAP use influenced the within participant change in CFA ln(IMT) from baseline to 12 months. These details are now given in the Statistical Analysis section (see Revised Manuscript Page 13, Lines 255-266) along with mention of diagnostic plots used to check model assumptions as explained in 1.2 and 1.3 above.

These results are reported and discussed (see Revised Manuscript Pages 24-25, Lines 405-426; Page 32, Lines 566-574) while Figure 3 illustrates these findings. 

1.5 Reviewer Question: “The R^2 are accompanied by a p-value. What does it refer to? Are you really comparing the intercept-only model with a model where the covariates are significant, and does obviuously improve the fit? In general, model fitting is rather poor, so there is a lot of unexplained heterogeneity.”

Response: The p-values accompanying R^2 values have been removed from Table 4, and supplementary Table S4. We agree that there is a lot of unexplained heterogeneity in our data set. Consequently, the relationships between IMT and both base and SDB variables, as detected in our statistical models, only partly explain (at most 21-27%) the overall IMT variance present in this relatively small cohort. We point this out in the Discussion (see Revised Manuscript Pages 27, 29 and 32).

Reviewer #2: “This is an interesting article investigating the relationship between IMT of CCA/CFA and polysomnographic parameters. The results showed no significant association between severity of sleep apnea and carotid IMT. However, small association was found between femoral IMT and AHI/RDI. Otherwise, CPAP therapy for 12 months did not alter carotid IMT but reduced femoral IMT that was correlated with weight change instead of CPAP use.”

2.1. Reviewer Question: “Some important variables such as mini-O2 and snoring can be associated with carotid IMT and need to be included for analyses.”

Response: We have included analyses for minimum O2 saturation for both the Main and RFM sub group (see Revised Manuscript Tables 1, 2, 3 and 5 and Supplemental Tables S3, 5 and S6). Snoring metrics were available for the RFM sub group only (see Revised Manuscript Table 2).

Neither of these variables were significant predictors for CCA or CFA IMT (see Revised Manuscript Table 5 and Supplemental Tables S6).

2.2 Reviewer Question: “The change of IMT in one year period could be very tiny. Thus persistent follow-up is necessary to clarify the long-term effect. Regarding CPAP outcome in terms of IMT, no progression is another explanation in comparison to progression of IMT in non-treatment group patients. This can be put into discussion or do subgroup analysis to clarify the CPAP effect.”

Response: As suggested we have done a sub-group analysis to clarify any CPAP effect. No progression for CCA IMT depended on whether the patient gained or lost weight across the 12 month RFM period (see Revised Manuscript Figure 3A). Consequently, the no progression for CCA IMT found the RFM sub group as a whole, is actually a reflection of reduced values for subjects with weight loss and increased values for those with weight gain. There was no interaction with CPAP use.

CFA IMT, however, decreased for the RFM sub-group across the 12 month intervention associated with a joint effect of weight loss and CPAP use (see Revised Manuscript Figure 3B). 

2.3.Reviewer Question: “The clinical meaning and significance of femoral IMT in contribution to OSA need discussion.”

Response: This is the first time this finding has been reported as an observational finding. The clinical significance is unknown and is not informed by our study. We note that there are literature reports suggesting a closer relationship between CFA IMT and systemic cardiovascular health metrics than has been reported for CCA IMT where hemodynamic factors may play a bigger role (see Revised Manuscript Page 5, Lines 80-84.). Whether the linkage between CFA IMT and sleep disordered breathing severity, as detected in our study, represents a more “systemic” pathophysiological interaction remains speculative.

We have expanded our discussion of potential links between SDB and CFA IMT on Page 31, Lines 560-562 of the Revised Manuscript.

Reviewer #3: “Review of the manuscript “Predictors for carotid and femoral artery intima-media thickness in a non diabetic sleep clinic cohort”

3.1 Reviewer Question: “The study is interesting, however, there are some main concerns to be evaluated. Changing several variables in the 12-month sub-study can leave the analyzes confusing. BMI and OSA severity are collinear variables. Even when conducting collinearity tests, with a relatively small number of patients, we may have erroneous conclusions. “

Response: We acknowledge the limitations of our relatively small sample size (see Revised Manuscript Page 32, Lines 576-586) but our RFM sub-group study is one of the larger and longer follow up studies reported. We acknowledge the potential for interactions across the 12 month intervention and this was the reasoning behind using linear mixed effects modelling to analyse the data. Indeed, this approach did reveal an interaction between weight loss and CPAP compliance for change in CFA IMT.

3.2 Reviewer Question: “Another fundamental point is the inclusion of smokers in this study Smoking increases cardiovascular risk by 20 x, so it is difficult to analyze risk factors of such different importance in a study of approx. 200 participants. Please insert more information about smoking in the text.”

Response: We included smoking as a risk factor on the basis of the participant reporting current smoking or having ever smoked or were currently smoking versus having never smoked. We thus combined current smokers and past smokers into one group. We have clarified our smoking classification in the Revised Manuscript on Page 8, Line 141). We used this approach because we had insufficient current smokers (n=11) to analyse separately (114 past smokers, 166 non-smokers, 5 missing data). Since past smoking history (which also may have been minimal and years ago) is not as strong a cardiovascular risk factor compared with current smoking, we suspect that we did not identify smoking history as a predictor of CCA or CFA IMT because there were very few (n=11) current smokers in our (relatively small) cohort. This point has been noted in the Discussion (see Revised Manuscript Page 26, Lines 433-435).

3.3 Reviewer Question : “Authors must inform cpap adherence data, which is essential for data interpretation. The SAVE Study was cited, but the main criticism of the study is precisely the low adherence to cpap. Therefore, a reduction in cardiovascular risk with cpap cannot be assessed without this analysis. The adherence group (> 4 h of daily use) vs the non-adherence group (<4 h of daily use) should be evaluated”.

Response: In response to the reviewer’s comment suggesting reanalysing the data for CPAP compliance of > 4 hr daily versus < 4 hr daily, we have now specifically incorporated this factor into the linear mixed effects modelling for the RFM sub group. There were no effects on CCA IMT over the 12 month intervention, but we can report a 12.9% (95%CI 6.8, 18.7%, p=0.001) reduction in CFA IMT from baseline in those who both lost weight and used CPAP >=4hours/day, but not in those who gained/maintained weight or used CPAP <4hours/day ( see Revised Manuscript Figure 3). Thus, using CPAP for >4 hours daily was associated with a decrease in CFA IMT, but only those in participants who also lost weight across the 12 month intervention. This finding is discussed in the Revised Manucript on Page 32, Lines 566-574.

3.4 Reviewer Question: “Describe whether fixed or automatic cpap was used. Describe the AIH after titration.”

Response: CPAP was fixed. For the RFM sub group (n=137) the mean AHI value (± SD) after titration was 1.7 ± 2.1 events/hr. We have now reported these data in the revised manuscript (see Revised Manuscript Page 24, Line 392).

---

## [Decision Letter · Decision Letter 1]

19 May 2021

Predictors for carotid and femoral artery intima-media thickness in a non-diabetic sleep clinic cohort

PONE-D-20-02731R1

Dear Dr. Amis,

We’re pleased to inform you that your manuscript has been judged scientifically suitable for publication and will be formally accepted for publication once it meets all outstanding technical requirements.

Kind regards,

Giuseppe Andò, M.D., Ph.D.

Academic Editor

PLOS ONE

Additional Editor Comments (optional):

Reviewers' comments:

Reviewer's Responses to Questions

**Comments to the Author**

1. If the authors have adequately addressed your comments raised in a previous round of review and you feel that this manuscript is now acceptable for publication, you may indicate that here to bypass the “Comments to the Author” section, enter your conflict of interest statement in the “Confidential to Editor” section, and submit your "Accept" recommendation.

Reviewer #1: All comments have been addressed

Reviewer #2: All comments have been addressed

2. Is the manuscript technically sound, and do the data support the conclusions?

Reviewer #1: (No Response)

Reviewer #2: Yes

3. Has the statistical analysis been performed appropriately and rigorously? 

Reviewer #1: (No Response)

Reviewer #2: Yes

4. Have the authors made all data underlying the findings in their manuscript fully available?

Reviewer #1: (No Response)

Reviewer #2: Yes

5. Is the manuscript presented in an intelligible fashion and written in standard English?

Reviewer #1: (No Response)

Reviewer #2: Yes

6. Review Comments to the Author

Reviewer #1: (No Response)

Reviewer #2: (No Response)

7. PLOS authors have the option to publish the peer review history of their article (what does this mean?). If published, this will include your full peer review and any attached files.

Reviewer #1: No

Reviewer #2: No

---

## [Editor Report · Acceptance letter]

24 May 2021

PONE-D-20-02731R1 

Predictors for carotid and femoral artery intima-media thickness in a non-diabetic sleep clinic cohort 

Dear Dr. Amis:

I'm pleased to inform you that your manuscript has been deemed suitable for publication in PLOS ONE. Congratulations! Your manuscript is now with our production department. 

Kind regards, 

on behalf of

Dr. Giuseppe Andò 

Academic Editor

PLOS ONE